# Knockdown of the *Halloween* Genes *spook*, *shadow* and *shade* Influences Oocyte Development, Egg Shape, Oviposition and Hatching in the Desert Locust

**DOI:** 10.3390/ijms23169232

**Published:** 2022-08-17

**Authors:** Sam Schellens, Cynthia Lenaerts, María del Rocío Pérez Baca, Dorien Cools, Paulien Peeters, Elisabeth Marchal, Jozef Vanden Broeck

**Affiliations:** 1Molecular Developmental Physiology and Signal Transduction, KU Leuven, Naamsestraat 59 Box 2465, B-3000 Leuven, Belgium; 2Center for Medical Genetics Ghent, UZ Ghent, Corneel Heymanslaan 10 Entrance 34, B-9000 Ghent, Belgium; 3Department of Life Science Technologies, Imec, Kapeldreef 75, B-3001 Leuven, Belgium

**Keywords:** biosynthesis, ecdysone, ecdysteriod, hemimetabola, hormone, insect, oogenesis, reproduction

## Abstract

Ecdysteroids are widely investigated for their role during the molting cascade in insects; however, they are also involved in the development of the female reproductive system. Ecdysteroids are synthesized from cholesterol, which is further converted via a series of enzymatic steps into the main molting hormone, 20-hydoxyecdysone. Most of these biosynthetic conversion steps involve the activity of cytochrome P450 (CYP) hydroxylases, which are encoded by the *Halloween* genes. Three of these genes, *spook* (*spo*), *phantom* (*phm*) and *shade* (*shd*), were previously characterized in the desert locust, *Schistocerca gregaria*. Based on recent sequencing data, we have now identified the sequences of *disembodied* (*dib*) and *shadow* (*sad*), for which we also analyzed spatiotemporal expression profiles using qRT-PCR. Furthermore, we investigated the possible role(s) of five different *Halloween* genes in the oogenesis process by means of RNA interference mediated knockdown experiments. Our results showed that depleting the expression of *SchgrSpo*, *SchgrSad* and *SchgrShd* had a significant impact on oocyte development, oviposition and hatching of the eggs. Moreover, the shape of the growing oocytes, as well as the deposited eggs, was very drastically altered by the experimental treatments. Consequently, it can be proposed that these three enzymes play an important role in oogenesis.

## 1. Introduction

Ecdysteroids are lipophilic polyhydroxylated hormones that are widely investigated for their regulatory role in the molting of insects. They initiate the process of replacing the old cuticle by a newer and larger one, which allows the insects to grow and to pass from one instar to the next. In larval insects, ecdysone (E) is produced in the glandular cells of the prothoracic glands (PG) and released in the hemolymph. In peripheral tissues, such as the Malpighian tubules, E will be converted into the active molting hormone, 20-hydroxyecdysone (20E). In adult insects, the prothoracic glands are known to degenerate and ecdysteroidogenesis is taken over by the reproductive organs. Furthermore, in addition to the prothoracic glands and the reproductive organs, some other sites, such as oenocytes and epidermis, may also function as sources of ecdysteroids in insects, as reviewed by Delbecque et al. (1990), Gilbert et al. (2002) and Lafont and Koolman (2009) [1,2,3].

The response to 20E is mediated at the site of the cell nucleus by a heterodimeric receptor complex composed of the ecdysone receptor (EcR) and the ultraspiracle/retinoid-X-receptor (USP/RXR) [4]. In general, the 20E bound complex interacts with ecdysone response elements (EcRE) and as such, may induce transcription of 20E target genes. First, the early genes, including the *ecdysone-induced protein 74* and *75* (*E74* and *E75*), as well as *Broad-Complex* (*Br-C*), will be transcribed [5,6,7,8]. With a slight delay, transcription of the early-late genes, including *E78*, *hormone receptor 3* and *39* (*HR3* and *HR39*), will take place [9,10,11] and, together with the early genes, these will induce the transcription of the late genes, such as *fushi tarazu transcription factor 1* (FTZ-F1) [12]. However, besides this 20E-dependent activation, the ligand-independent activity of these nuclear receptors has recently been described in *Drosophila* [13].

In contrast to vertebrates, insects are not able to synthesize cholesterol, the biosynthetic precursor of ecdysteroids, from single carbon molecules, but need to obtain sterols from their diet. Already in the intestine of the insects, plant-derived phytosterols will be dealkylated to cholesterol [2]. In the microsomes of a cell, cholesterol will be further converted into 7-dehydrocholesterol (7dC), which is then released into the cytosol and converted into diketol. The exact sequence of enzymatic reactions involved in this conversion is unknown and described as the ‘Black Box’ in ecdysteroidogenesis. One rate-limiting cytochrome P450 (CYP) enzyme in this ‘Black Box’ is encoded by the *Halloween* gene *spook* (*spo*) [14,15]. When returned into the microsomes, diketol will be hydroxylated into ketodiol, which, at its turn, is hydroxylated to ketotriol by a 25-hydroxylase [16,17]. In the mitochondria of the cell, ketotriol will be hydroxylated to 2-deoxyecdysone (2dE) by a 22-hydroxylase, and 2dE will be further hydroxylated to E by a 2-hydroxylase [18,19]. These hydroxylation steps are catalyzed by cytochrome P450 (CYP) enzymes that are encoded by the *Halloween* genes *phantom* (*phm*), *disembodied* (*dib*) and *shadow* (*sad*), respectively [16,17,18,19]. Finally, E will be released in the hemolymph and in peripheral tissues hydroxylated to 20E by a 20-hydroxylase which is encoded by the *Halloween* gene *shade* (*shd*) [20]. For an extensive overview of the ecdysteroid biosynthesis pathway, the reader is referred to the following reviews: Gilbert and Warren (2005), Lafont (2005) and Rewitz et al. (2006) [21,22,23].

Together with juvenile hormones (JHs), ecdysteroids are considered as the classic insect hormones involved in oogenesis of insects; however, their function and importance differ between distinct insect orders [24]. The dominant role of ecdysteroids in the oogenesis process in higher Diptera is widely investigated in the fruit fly, *Drosophila melanogaster*. Ecdysteroids are needed to establish and maintain the stem cell niche [25], to stimulate follicle cell formation and border cell migration [26], to coordinate the onset of vitellogenesis with the availability of nutrients [27], to stimulate yolk polypeptide synthesis in the fat body [28,29], and finally, to induce choriogenesis [30] (for an extensive overview of these different roles of ecdysteroids, see Belles and Piulachs (2014) and Swevers (2018)) [31,32]. In contrast to this extensive knowledge on the role of ecdysteroids in higher Diptera, their function in insects that use JH as the primary regulator of ovarian maturation (such as cockroaches and locusts) is less documented. Reports have shown that the ovaries of the migratory locust, *Locusta migratoria,* produce large amounts of ecdysteroids at the end of the gonadotrophic cycle. These ecdysteroids are mainly incorporated in the terminal oocytes as a maternal source for embryogenesis [33]. Ecdysteroids have also been described to induce meiotic re-initiation in this same species, thereby stimulating ovum maturation [34]. In the German cockroach, *Blattella germanica*, 20E was found to induce the start of the choriogenesis process when vitellogenesis is completed [35,36]. More recently, we have silenced EcR and RXR in the desert locust, *Schistocerca gregaria*, and have also shown that these ecdysteroid receptor components are essential for normal choriogenesis [37]. In this species, transcript levels of *SchgrSpo* and *SchgrPhm* were found to rise at the end of the first gonadotrophic cycle, in accordance with the accumulation of ecdysteroids in oocytes and their role in choriogenesis [38,39]. Our current study provides further transcript profiling of *SchgrSpo* and *SchgrPhm* and three additional Halloween genes (*SchgrDib*, *SchgrSad* and *SchgrShd*) in *S. gregaria* oogenesis. RNAi-mediated knockdown of *SchgrSpo*, *SchgrSad* and *SchgrShd* genes significantly affected oocyte development, oviposition and hatching of the eggs, suggesting that they play an important role in the female reproductive physiology of the desert locust, an insect species in which JH is known to be the main regulator.

## 2. Results

### 2.1. Sequence Analysis of SchgrDib and SchgrSad

The complete *SchgrDib* and *SchgrSad* open reading frame (ORF) sequences were found in in-house *S. gregaria* transcriptome and recently published *S. gregaria* genome databases (SCHGR_00006992 and SCHGR_00008835, respectively) (Appendix A) [40]. The *SchgrDib* ORF (GenBank acc. no. KY404120) comprises 1578 nucleotides encoding a predicted protein of 526 amino acids, whereas the *SchgrSad* ORF (GenBank acc. No. MZ780957) comprises 1443 nucleotides encoding a predicted protein of 481 amino acids. These amino acid sequences are included in a multiple sequence alignment with previously identified insect DIB (Appendix A) or SAD proteins (Appendix A) [19,41,42,43,44]. *SchgrDib* and *SchgrSad* displayed 47.5% and 40.7% identity with their respective functionally characterized orthologs in *D. melanogaster*. Furthermore, several key motifs for CYP450 enzymes, namely, the mitochondrial signal sequence, the P/G rich domain, helix-C, helix-I, helix-K, the PERF-motif, and the heme-binding domain, could be identified (Appendix A) [45].

### 2.2. Tissue Distribution and Developmental Transcript Profile during the First Gonadotrophic Cycle

Tissue and temporal distribution profiles of *SchgrDib*, *SchgrSad* and *SchgrShd* transcripts are shown in Figure 1, whereas transcript profiles of *SchgrSpo* and *SchgrPhm* were previously published by Marchal et al. (2011) [39]. Transcript levels of the *Halloween* genes *SchgrDib*, *SchgrSad* and *SchgrShd* were determined in the brain, *corpora cardiaca*, suboesophageal ganglion, prothoracic glands, thoracic ganglia, fat body, midgut and ovaries of virgin females and the testes and accessory glands of virgin males, both 10 days after the final molt, using qRT-PCR (Figure 1A,C,E). Transcript levels of *SchgrDib* were found to be significantly higher in the ovaries, while *SchgrSad* transcript levels were found to be significantly higher in the prothoracic glands of female adult locusts when compared to other tissues (Figure 1A,C). *SchgrShd* on the other hand, showed a wide distribution profile with high transcript levels in the ovaries, suboesophageal ganglion and brains of female adult locusts and testes and accessory glands of male adult locusts (Figure 1E). Furthermore, the temporal distribution profiles of *SchgrDib*, *SchgrSad* and *SchgrShd* were determined in the ovaries throughout the first reproductive cycle (Figure 1B,D,F). *SchgrDib* transcript levels were the highest on the day of molting to the adult stage; throughout the rest of the first reproductive cycle, the transcript levels were lower and remained relatively stable, showing no correlation with the ecdysteroid titer (Figure 1B). In contrast to *SchgrDib* transcript levels, the temporal expression of *SchgrSad* and *SchgrShd* was increased towards the end of reproductive cycle, correlating with the observed peak in ecdysteroid titer (Figure 1D,F).

### 2.3. RNA Interference of the Individual Halloween Genes

To investigate the role of ecdysteroids in the female reproductive physiology of *S. gregaria*, the locust orthologs of five ecdysteroid biosynthesis genes were silenced using RNAi. Female locusts were injected with dsRNA targeting *SchgrSpo* (*dsSpo)*, *SchgrPhm* (*dsPhm*), *SchgrDib* (*dsDib*), *SchgrSad* (*dsSad*) or *SchgrShd* (*dsShd*) on day 6 of the 5th nymphal stage (N5D6), day 1 (AdD1), day 5 (AdD5) and day 9 (AdD9) of the adult stage. Control locusts were injected with *dsGFP* following the same injection scheme. First, one group of locusts was sacrificed twelve days after molting to the adult stage (AdD12) to investigate the possible effects of the RNAi-mediated knockdown on oocyte development and on the transcript levels of several genes of interest. Second, a group of locusts was kept alive to investigate the possible effects of the RNAi treatment on mating, fecundity and fertility (Appendix A).

#### 2.3.1. Knockdown Efficiency and Effect on Transcript Levels of Target Genes

Transcript levels of the *Halloween* genes were investigated in the dissected ovaries of female locusts twelve days after the final molt using qRT-PCR. Analysis of the normalized Ct-values indicated that the levels of *SchgrSpo*, *SchgrPhm*, *SchgrDib*, *SchgrSad* and *SchgrShd* transcripts were significantly reduced in their respective knockdown conditions (*dsSpo*, *dsPhm*, *dsDib*, *dsSad* and *dsShd*) with 93%, 88%, 46%, 98% and 98%, respectively (*p* = 0.0234, *p* = 0.0036, *p* = 0.0237, *p* = 0.0130, *p* = 0.0043, respectively; two-sided unpaired *t*-test on log-transformed data including Welch’s correction for comparison of *SchgrSad* transcript levels between *dsGFP* and *dsSad*) (Appendix A). Furthermore, *SchgrPhm* transcript levels were significantly lower in the *dsSpo*- and *dsSad*- treated females (69%; *p* = 0.0413 and 74%; *p* = 0.0299, respectively; two-sided Unpaired *t*-test on log-transformed data) and *SchgrDib* transcript levels in *dsShd*- treated females (42%; *p* = 0.0253; two-sided Unpaired *t*-test on log-transformed data) when results were compared to the control (*dsGFP*) condition (Appendix A).

The effect of knocking down the ecdysteroid biosynthesis gene orthologs on JH signalling and vitellogenin synthesis was investigated by measuring the transcript levels of the JH receptor gene *methoprene-tolerant* (*SchgrMet*) and the JH response gene *Krüppel-homolog 1* (*SchgrKr-h1*) in the fat body and ovaries, as well as the transcript levels of *vitellogenin 1* and *vitellogenin 2* (*SchgrVg1* and *SchgrVg2*) in the fat body, of all experimental animals. However, no significant differences in expression of these genes were observed when compared with *dsGFP*-injected (control) females (Appendix A).

#### 2.3.2. Observations of Oocyte Size, Mating, Oviposition and Hatching

Analysis of the average length and width of basal oocytes in fifteen female locusts, dissected on AdD12, per condition showed that the oocytes of *dsSpo*-, *dsSad*-, *dsShd*-treated females had an abnormal, more spherical shape when compared to the typical ovoid shape of the oocytes of *dsGFP*-treated females (Figure 2A–C + Appendix A). Moreover, basal oocytes in these three conditions were significantly shorter than oocytes derived from control animals (*dsSpo p* < 0.0001; *dsSad p* = 0.0008; *dsShd p* = 0.0059; One-way ANOVA with Dunnett’s Multiple Comparisons Test; Figure 2D), whereas their width did not significantly differ between any of the tested conditions (One-way ANOVA with Dunnett’s Multiple Comparisons Test; Figure 2E). Furthermore, plotting the average length in function of the average width showed that oocytes of *dsSpo*-, *dsSad*-, *dsShd*-treated females were significantly shorter than control oocytes at a given width, whereas estimations of the oocyte volume did not reveal any significant differences between the knockdown and control conditions (Kruskal–Wallis test with Dunn’s multiple comparison test; Appendix A). Knockdown of *SchgrPhm* and *SchgrDib* did not result in this phenotype (Figure 2F). Confocal imaging of DAPI-stained follicles did not reveal any obvious structural differences in the follicular epithelium between the various knockdown (*dsSpo*, *dsPhm*, *dsDib*, *dsSad* and *dsShd*-injected) conditions and the *dsGFP*-injected control (Appendix A).

Although no significant differences in the occurrence of mating were observed between any of the tested conditions (Appendix A), the cumulative percentage of ovipositing females over time significantly differed between *dsGFP*- (control) (83.33%) and *dsSpo*- (65%) or *dsSad*- (50%) injected females (*p* = 0.0335 and *p* = 0.0448, respectively; Mantel–Cox test; Figure 3A). Nevertheless, the average number of days between copulation and egg laying, as well as the number of laid eggs, did not differ significantly between any of the tested conditions (Figure 3B,C). However, in accordance with the oocyte measurements, the eggs deposited by *dsSpo*-, *dsSad*-, *dsShd*-treated females were significantly shorter (*p* = 0.0028, *p* = 0.0004 and *p* = 0.0003, respectively; Kruskal–Wallis test with Dunn’s Multiple Comparisons Test) and eggs from *dsSpo*-treated females were also significantly wider than eggs derived from *dsGFP*-treated females (*p* = 0.0364; Kruskal–Wallis test with Dunn’s Multiple Comparisons Test; Figure 3D–F). Furthermore, the estimated volume of the eggs was not significantly affected in the knockdown conditions when compared to the control condition (One-way ANOVA with Dunnett’s multiple comparison test; Appendix A). Observational analysis of hatching showed that the average number of days between egg laying and hatching did not significantly differ between any of the tested conditions (Figure 3G), that the average number of hatchlings derived from *dsSpo*-, *dsSad*- and *dsShd*-treated females was significantly lower (*p* = 0.0012, *p* = 0.0079 and *p* = 0.0286, respectively; One-way ANOVA with Dunnett’s Multiple Comparisons Test; Figure 3H) and that *dsSpo*- and *dsSad*-treatment conditions had a correspondingly lower percentage of hatching success (=100 × number of hatchlings/number of eggs laid; *p* = 0.0021 and *p* = 0.0103, respectively; Kruskal–Wallis test with Dunn’s Multiple Comparisons Test; Figure 3I), when compared to *dsGFP*-treated females.

### 2.4. RNA Interference of Halloween Gene Combinations

Since injections of *dsSpo*, *dsSad* and *dsShd* resulted in females containing significantly shorter, more spherical, basal oocytes than *dsGFP*-injected (control) females, while neither *dsPhm* nor *dsDib* injections revealed this phenotype, two combinations of *Halloween* genes were targeted, i.e., *SchgrSpo*, *SchgrSad* and *SchgrShd* (*dsSpo*/*Sad*/*Shd*) and *SchgrPhm* and *SchgrDib* (*dsPhm*/*Dib*) in a follow-up experiment. The *dsSpo*/*Sad*/*Shd* knockdown condition was selected to verify whether an even more pronounced effect would be generated, whereas the *dsPhm*/*Dib* condition was included to find out whether the spherical oocyte/egg phenotype would perhaps appear upon a combined knockdown. Similarly, as for the separate gene knockdown experiment, a first group of injected locusts was used to investigate possible effects on oocyte development and on the transcript levels of several genes of interest, whereas a second group of locusts was kept alive to enable us to observe mating, egg laying and hatching.

#### 2.4.1. Knockdown Efficiency and Effect on Transcript Levels of Target Genes

qRT-PCR analysis of the *Halloween* gene transcript levels in 12-day adult females revealed that *SchgrPhm* (66%) and *SchgrShd* (81%) were significantly downregulated in the *dsSpo*/*Sad*/*Shd* knockdown condition (*p* < 0.0001 for both transcripts; two-sided Unpaired *t*-test on log-transformed data including the Welch’s correction for the comparison of the *SchgrShd* transcript levels) and *SchgrPhm* (90%) and *SchgrDib* (60%) were significantly downregulated in the *dsPhm*/*Dib* knockdown condition (*p* < 0.0001 for both transcripts; two-sided Unpaired *t*-test on log-transformed data; Appendix A). Additionally, no significant differences were observed for *SchgrMet* and *SchgrKr*-*h1* transcript levels in fat body and ovaries, as well as *SchgrVg1* and *SchgrVg2* transcript levels in fat body of *dsSpo*/*Sad*/*Shd*- and *dsPhm*/*Dib*-treated females, when compared with *dsGFP*-injected control females (Appendix A).

#### 2.4.2. Observations of Oocyte Size, Mating, Oviposition and Hatching

The analysis of oocyte length and width indicated that basal oocytes of *dsSpo*/*Sad*/*Shd*-treated females were significantly shorter than oocytes of the *dsGFP*-treated females (*p* < 0.0001; Kruskal–Wallis test with Dunn’s Multiple Comparisons Test; Figure 4A). However, the average oocyte width of *dsSpo*/*Sad*/*Shd*-treated females did not significantly differ from the control condition, meaning that, in line with the knockdown of the individual *Halloween* genes, oocytes of the combined knockdown condition were significantly shorter than control oocytes at a certain width (Figure 4B,C). Furthermore, their estimated oocyte volume did not significantly differ from the estimated oocyte volume of the control condition (Kruskal–Wallis test with Dunn’s multiple comparison test; Appendix A). Consequently, oocytes from *dsSpo*/*Sad*/*Shd*-treated females had a more spherical shape when compared to the ovoid shape of oocytes from control females (Appendix A). In contrast to the findings for *dsPhm*- and *dsDib*-treated females, basal oocytes of females treated with *dsPhm*/*Dib* were significantly shorter and thinner than basal oocytes from *dsGFP*-injected (control) females (*p* = 0.0072 and *p* = 0.0155, respectively; Kruskal–Wallis test with Dunn’s Multiple Comparisons Test; Figure 4A–C; Appendix A). As a consequence, these oocytes were also found to have a significantly smaller estimated volume than in the control condition (*p* = 0.0126; Kruskal–Wallis test with Dunn’s multiple comparison test; Appendix A). Furthermore, no major differences were observed in DAPI-stained follicular epithelia between knockdown (*dsSpo*/*Sad*/*Shd* and *dsPhm*/*Dib*) and control (*dsGFP*) conditions (Appendix A).

Despite the observed effects on oocyte size, observations of mating revealed no significant differences between any of the tested conditions (Appendix A). However, only 19% of the *dsSpo*/*Sad*/*Shd*-treated females (4 out of 21) were able to lay eggs over a period of 45 days (Figure 5A). This result is in huge contrast with the 87.5% and 96% which were observed for the *dsPhm*/*Dib*- and *dsGFP*-treated females, respectively. Although most females of the *dsPhm*/*Dib* condition were able to lay their eggs, it took them significantly more time in comparison to *dsGFP*-injected (control) females (*p* = 0.0187; Kruskal–Wallis test with Dunn’s Multiple Comparisons Test; Figure 5B). Moreover, females from the *dsPhm*/*Dib* knockdown condition were found to lay significantly more eggs than females from the control condition (*p* = 0.0134; Kruskal–Wallis test with Dunn’s Multiple Comparisons Test; Figure 5C). Measurements of the egg length and width indicated that eggs derived from both knockdown conditions, *dsSpo*/*Sad*/*Shd* and *dsPhm*/*Dib*, were significantly shorter than these derived from *dsGFP*-injected (control) females (*p* = 0.0027 and *p* = 0.0019, respectively; Kruskal–Wallis test with Dunn’s Multiple Comparisons Test), whereas their width was not significantly affected (Kruskal–Wallis test with Dunn’s multiple comparisons test; Figure 5D–F). Furthermore, the estimated volume of the eggs did not significantly differ between all conditions (Kruskal–Wallis test with Dunn’s multiple comparison test; Appendix A). Besides the significant effect observed on egg laying, eggs of the *dsPhm*/*Dib* knockdown condition took significantly more time to hatch and, although the number of hatchlings was not significantly different, the hatching success was significantly lower when compared to the *dsGFP* control condition (*p* = 0.0028, *p* = 0.8076 and *p* = 0.0188, respectively; Kruskal–Wallis test with Dunn’s Multiple Comparisons Test; Figure 5G–I). The average number of days between egg laying and hatching, the average number of hatchlings and the average hatching success (=100 × number of hatchlings/number of eggs laid) were not affected significantly after simultaneously targeting *SchgrSpo*, *SchgrSad* and *SchgrShd*.

## 3. Discussion

### 3.1. Characteristics and Expression Patterns of SchgrDib, SchgrSad and SchgrShd

In the current study, the nucleotide sequences of the *Halloween* genes *Disembodied* (CYP302A1, *SchgrDib*) and *Shadow* (CYP315a1, *SchgrSad*) in the desert locust, *S. gregaria*, were described (Appendix A). Based on their amino acid sequence similarities with other known DIB and SAD proteins, several key motifs of cytochrome P450 (CYP450) enzymes were identified (Appendix A) [19,41,42,43,44]. Earlier phylogenetic studies have shown that DIB and SAD belong to a monophyletic group of mitochondrial P450s [47], exhibiting common characteristics, such as charged residues in the N-terminal target sequence and two positively charged residues near the heme-binding domain that facilitate redox partner interactions [45,48]. DIB was identified in *D. melanogaster* and *B. mori* as a C22-hydroxylase, converting 2,22-dideoxyecdysone to 2-deoxyedysone, whereas SAD was identified as a C2-hydroxylase, converting 2-deoxyecdysone to ecdysone [19,42]. Recently, Dib and Sad orthologs were identified in *L. migratoria*, and based on their conserved phylogenetic relationship with representative insects, it was suggested that both enzymes are involved in the synthesis of the molting hormone [41]. Since *SchgrDib* and *SchgrSad* have 89% and 81% sequence identity with their respective orthologs in *L migratoria* and since they show 48% and 41% sequence identity with the functionally conserved orthologs in *D. melanogaster* [19], it is likely that both enzymes are also part of the ecdysteroidogenesis pathway in the desert locust.

Tissue and temporal distribution profiles of the *Halloween* genes *SchgrSpo* and *SchgrPhm* in adult locusts were previously published by Marchal et al. [39]. *SchgrSpo* and *SchgrPhm* relative transcript levels were highest in the prothoracic glands and in the ovaries of adult female locusts. Similarly, transcript levels of *SchgrSad* were highest in the prothoracic glands and high relative transcript levels of *SchgrDib* and *SchgrShd* were found in the ovaries (Figure 1A,C,E). Contrary to most insect species, the prothoracic glands still persist in adult locusts after the final molt, but they do not release ecdysteroids anymore [49]. Therefore, the functional importance of the *Halloween* gene expression in the adult prothoracic glands is still unclear. The expression of the *Halloween* gene transcript levels in the ovaries is consistent with ecdysteroid synthesis in the adult ovaries as has been described in Diptera and Lepidoptera [50,51,52]. More recently, research in a hemipteran species, *Nilaparvata lugens*, and an orthopteran species, *L. migratoria*, also showed the relatively high transcript levels of *Dib* and *Shd* in the ovaries of female adults [41,53]. Contrary to temporal distributions of the *Halloween* genes in *A. aegypti* and the previously published distributions of *SchgrSpo* and *SchgrPhm* [39,52], no correlation was observed between *SchgrDib* transcript levels and the circulating ecdysteroid titer in the adult female *S. gregaria*. On the other hand, transcript levels of *SchgrSad* and *SchgrShd* increased during the female reproductive cycle, which coincided with the observed peak in ecdysteroid titer (Figure 1B,D,F). However, a high variation could be observed at these time points, suggesting that transcription of these genes may be tightly regulated, showing dynamic pulses of transcript levels when induced. A similar pattern was seen for the transcript levels of *BlageShd* in the ovaries of the German cockroach *B. germanica* [54]. In the brown planthopper, *N. lugens*, it was suggested that both *Nillu*CYP307B1 (Spo) and *Nillu*CYP314A (Shd) are rate-limiting enzymes for ecdysteroidogenesis [53]. Therefore, it is evident that the expression profile of these genes correlates well with the circulating ecdysteroid titer [14,45]. Since DIB does not act as rate-limiting enzyme, its transcript levels are probably less strictly regulated.

### 3.2. Halloween Gene Expression Is Crucial in Female S. gregaria Reproductive Physiology Affecting Egg Shape, Oviposition and Hatching

Both experimental conditions, knocking down the *Halloween* genes *SchgrSpo*, *SchgrSad* and *SchgrShd,* separately (*dsSpo*, *dsSad* and *dsShd*) or in combination (*dsSpo*/*Sad*/*Shd*), led to the observation of abnormally shaped, more spherical oocytes that were significantly shorter in length than the control condition, while their width and estimated volume were not reduced (Figure 2 and Figure 4; Appendix A). A very similar phenotype was seen in *B. germanica* and *T. castaneum* after depletion of Notch [55,56]. The Notch pathway is an essential regulator of cell proliferation and differentiation and, as such, is involved in determining cell fate [57,58]. In the panoistic ovary of *B. germanica*, Notch was found to play a role in inducing ovarian follicle elongation. Furthermore, also in three holometabolan insect species, *D. melanogaster*, *A. mellifera* and *T. castaneum*, the Notch pathway was found to play an important role in oogenesis [56,58,59,60]. Interestingly, in 2018, Yatsenko and Shcherbata identified a coordinated role between ecdysone signalling and Notch signaling in germline niche formation in *Drosophila* [61]. Furtermore, Ramos et al. (2020) suggested that the Notch downstream component Eyes Absent affects the ecdysteroidogenic pathway in *B. germanica* [54]. Therefore, it is possible that a depletion of *Halloween* gene expression in *S. gregaria* might also functionally interact with the Notch signaling pathway and, consequently, disturb the oogenesis process.

To verify if the more spherical oocyte phenotype observed after injections of female locusts with *dsSpo*, *dsSad, dsShd* or *dsSpo*/*Sad*/*Shd* might be correlated with disruption of the follicular epithelium, we have visualized DAPI-stained ovarian follicles by confocal microscopy. However, comparison with follicles derived from *dsGFP*-injected control females did not reveal any obvious structural differences in the follicular epithelial layer, which retained its normal appearance. Therefore, although differences at submicroscopic or molecular levels cannot be excluded, the more spherical oocyte shape does not appear to be associated with any obvious abnormalities in the surrounding epithelium.

Although the observed effect on egg laying in the *dsSpo*/*Sad*/*Shd*-treated females was more severe than the effect after treatment with *dsSpo, dsSad* or *dsShd* separately, qPCR-based transcript analysis revealed that only the *SchgrShd* transcript levels were significantly reduced (Appendix A). This may be due to compensatory mechanisms, similarly as previously reported for other *Halloween* gene knockdown experiments in *S. gregaria* [39] and in other organisms [62,63,64]. Furthermore, this qPCR analysis only represents one single timepoint at the end of the experiment when the locusts were sacrificed.

In contrast to the observed phenotypes in *dsSpo*-, *dsSad*, *dsShd*- and *dsSpo*/*Sad*/*Shd*-treated females, knockdown of *SchgrPhm* or *SchgrDib* separately did not reveal any effect on oocyte development, even though their transcript levels were significantly downregulated (Figure 2, Appendix A). However, downregulating both transcripts at the same time (*dsPhm*/*Dib*) resulted in a delayed oocyte development which was represented by oocytes that were significantly smaller in length and width than the ones from the control condition (Figure 4). Additionally, a significantly increased number of days between copulation and egg laying, as well as a significantly increased number of laid eggs, was observed for these *dsPhm*/*Dib*-treated females in comparison to *dsGFP*-treated females (Figure 5). It is generally known that female insects often exhibit a negative correlation between the number and the size of their eggs, constituting a trade-off between the offspring size and the available reproductive resources [65]. More specifically, in the desert locust, it was recently demonstrated that egg size and egg number may vary depending on their density-dependent phase state, having smaller but more eggs in isolated locusts when compared to crowded ones [66]. Moreover, already in 1999, it was shown that ovaries of solitarious *S. gregaria* females contain less ecdysteroids than gregarious ones [67]. Taken together, it might be proposed that the reproductive trade-off between solitarious and gregarious ovaries is correlated with the ecdysteroid content of these ovaries. Consequently, it can be suggested that knockdown of the *Halloween* genes, which is expected to lead to reduced ecdysteroid levels in the ovaries, may trigger changes in oocyte/egg size and number that are in line with differences that were previously reported between population density dependent locust phases.

Besides their effect on oocyte development, treatment with *dsSpo*, *dsSad* and *dsSpo*/*Sad*/*Shd* also affected egg laying, having significantly fewer females that successfully deposited their eggs than in the control condition (Figure 3 and Figure 5). These results are in accordance with the results obtained after knocking down the nuclear receptor complex (*SchgrECR*/*SchgrRXR*) in the same insect species which resulted in severely impaired oviposition [37]. Furthermore, significantly fewer eggs from the *dsSpo*-, *dsSad*- and *dsPhm*/*Dib*-treated females were hatching, in comparison to the eggs of the *dsGFP*-treated females (hatching success; Figure 3 and Figure 5). Since ecdysteroids are known to be incorporated into the developing oocytes to support embryogenesis [33], it could be suggested that knockdown of the *Halloween* genes may have resulted in insufficient ecdysteroid accumulation, hence disturbing the development of the embryo.

Knockdown of different *S. gregaria Halloween* genes did not always result in a very similar phenotypic outcome. Such variations were also seen in *T. castaneum* females, where knockdown of *TricaShd* resulted in arrested oocyte maturation, whereas knockdown of *TricaPhm* did not [68]. These discrepancies in the results for different genes acting in the same biosynthetic pathway could possibly be explained by the fact that Spo and Shd are known as important rate-limiting enzymes in the ecdysteroid biosynthesis pathway in different insect species [14,53]. Therefore, it could be suggested that the partial and transient depletion of *SchgrPhm* and *SchgrDib* (which is inherent to knockdown strategies) may have allowed for the synthesis of a level of ecdysteroids that was still sufficient for normal oocyte development. Another possible explanation could be that, in addition to E or 20E, one or more other ecdysteroids might also be capable of mediating this process. Interestingly, in some arthropod species, such as the spider mite, *Tetranychus urticae*, the Phm enzyme is lacking, and as a consequence, Ponasterone A is acting as the main ecdysteroid hormone [69]. In addition, since it is well known that multiple ecdysteroids are incorporated in the developing oocytes of different insect species in a conjugated form [70,71,72], it is possible that some of these conjugates, and hence, some hydroxylation steps, could be more important than others. Finally, it can also not be fully excluded that some enzymes encoded by *Halloween* genes might still exert as yet unknown activities in addition to their canonical role in the ecdysteroidogenesis process. However, since the knockdowns of three different enzymes, each responsible for a distinct step in the ecdysteroid biosynthesis pathway, have resulted in an identical phenotype, the latter option seems less likely.

### 3.3. Cross-Talk between Ecdysteroids and Other Hormonal Pathways

Multiple reports in different insect species proposed that cross-communication between JH and ecdysteroid signaling is taking place during the oogenesis process [36,70,73,74,75]. To further investigate this, the effect of knocking down the *Halloween* genes on transcript levels of *SchgrMet* and *SchgrKr*-*h1* was investigated using qRT-PCR. Both genes were previously indicated to be a good measure for the activity of JH [76,77,78,79]. Furthermore, transcript levels of *SchgrVg1* and *SchgrVg2* after knockdown of the different *Halloween* genes, separately and in combination, were investigated to evaluate the effect on vitellogenin synthesis. Other than expected, none of the tested knockdown conditions has induced any significant effect on *SchgrVg1* and *SchgrVg2* in the fat body and on *SchgrMet* or *SchgrKr-h1* expression levels in fat body or ovaries of females twelve days after the final molt (Appendix A). Nevertheless, previously published results from our lab showed that downregulating the nuclear receptor (*SchgrECR*/*SchgrRXR)* complex resulted in reduced *SchgrMet* transcript levels in the fat body [37]. These results were in accordance with results obtained in the cockroaches *Diploptera punctata* and *B. germanica*, suggesting that ecdysteroid signaling is involved in terminating the reproductive cycle [36,74]. However, why knockdown of *Halloween* genes did not result in a similar effect and how the interaction between ecdysteroid and JH signaling is regulated remains elusive. Perhaps, the effect of ECR/RXR on the JH signaling pathway might be ligand-independent, as this receptor complex was also found to prevent cell death in the PG of *Drosophila* in a ligand-independent manner [13].

Similarly to the results obtained after knockdown of *SchgrECR*/*SchgrRXR*, *vitellogenin* transcript levels were not significantly influenced by the *Halloween* gene RNAi treatments. These results indicate that, contrary to what had been suggested previously by J. Girardie and A. Girardie (1998), vitellogenin synthesis in locusts is most probably not directly regulated by ecdysteroids, while it is clearly dependent on JH [24].

The present study has shown that RNAi-mediated knockdown of the *Halloween* genes *SchgrSpo*, *SchgrSad* and *SchgrShd*, which code for ecdysteroid biosynthesis enzymes, had a clear effect on the development and shape of growing oocytes in adult female locusts, as well as on the shape (length/width ratio) of the deposited eggs. Moreover, oviposition and hatching success were negatively affected by this knockdown. Future research may shed more light on the functional interactions between ecdysteroid and JH signaling in the oogenesis process of female *S. gregaria* locusts.

## 4. Materials and Methods

### 4.1. Rearing of Animals

Desert locusts (*S. gregaria*) were reared under crowded conditions at a temperature of 30 ± 1 °C, 40–60% relative humidity and a 14/10 light/dark cycle. They were fed ad libitum with fresh cabbage leaves and rolled oats. Mature females in breeding cages were allowed to deposit their eggs in pots filled with a moistened turf/sand (2:1) mixture, which were collected and replaced on a weekly basis. Collected egg pots were placed in new empty cages, resulting in groups of hatchlings that differed no more than seven days in age. The adult locusts that were used for investigating the tissue distribution and temporal expression profiles of *SchgrDib*, *SchgrSad* and *SchgrShd* were derived from cages in which locusts were collected that all had molted to adulthood on the same day.

### 4.2. Sequence Analysis of SchgrDib and SchgrSad

The nucleotide sequences coding for the full-length *S. gregaria* orthologs of *dib* (KY404120) and *sad* (MZ780957) were retrieved from the recently published *S. gregaria* genome database [40], and also confirmed by an in-house transcriptome database (unpublished data). Using the M-Coffee web server in the default settings (http://tcoffee.crg.cat/apps/tcoffee/do:mcoffee; page accessed on 7 July 2020) [80], both sequences were aligned to orthologous sequences from four representative insect species (*Locusta migratoria*, *Drosophila melanogaster*, *Bombyx mori* and *Anopheles gambiae*) and conserved residues were highlighted using Boxshade (https://embnet.vital-it.ch/software/BOX_form.html; page accessed on 7 July 2020) [19,41,42,43,44].

### 4.3. Tissue Collection and RNA Extraction

To study the tissue distribution of *SchgrDib*, *SchgrSad* and *SchgrShd*, adult locusts were collected on their day of molting from the 5th nymphal stage to the adult stage, and ten days after this final molt (AdD10), the following tissues were dissected under a binocular microscope and rinsed in locust Ringer solution (150 mM NaCl, 1.7 mM CaCl_2_, 10 mM KCl, 4.3 mM MgCl_2_, 4 mM NaHCO_3_, 90 mM Sucrose, 5 mM Trehalose, pH 7.2): brain, *corpora cardiaca*, suboesophageal ganglion, prothoracic glands, thoracic ganglia, fat body, midgut and ovaries of virgin females, as well as testes and accessory glands of virgin males. These tissues were collected in three independent pools, each containing samples derived from 10 individual locusts, and were transferred to MagNA Lyser Green Beads containing Tubes (Roche) or RNase-free Screw Cap Microcentrifuge tubes.

To study the temporal transcript expression profiles of *SchgrDib*, *SchgrSad* and *SchgrShd*, groups of locusts that molted to adulthood on the same day were collected and ovaries of female locusts were dissected every other day, until day 18 of the adult stage. Samples were rinsed in locust Ringer solution and collected in MagNA Lyser Green Beads containing Tubes (Roche) in three independent pools, each containing six samples of individual locusts.

During the described RNAi experiments (see Section 4.6), virgin female locusts were dissected under a binocular microscope twelve days after the final molt. Tissues of interest (Ovaries and Fat body) were rinsed in locust Ringer solution and transferred to MagNA Lyser Green Beads containing Tubes (Roche). Samples derived from three individual locusts were pooled, generating five pools from a total of 15 female locusts. All samples were snap-frozen in liquid nitrogen and stored at −80 °C until further processing.

According to the different tissue collection strategies, different RNA extraction methods were used for larger, fat containing tissues and relatively small tissues containing low RNA quantities. Tissues collected in MagNA Lyser Green Bead Tubes (Roche) were homogenized in Qiazol (Qiagen) using the MagNa Lyser instrument according to the manufacturer’s instructions (1 min, 6500 rpm; Roche) and total RNA was extracted using the Qiagen^®^ RNeasy Lipid Tissue Kit, performing the optional DNase digestion protocol (RNase-Free DNase set, Qiagen, Hilden Germany). From smaller tissues collected in RNase-free Screw Cap Microcentrifuge tubes, total RNA was extracted using the RNAqueous-Micro Kit (Ambion) according to the manufacturer’s protocol, including the recommended DNase step. The purity and concentration of the RNA samples was analyzed using a Nanodrop spectrophotometer (Nanodrop ND-1000, Thermo Fisher Scientific, Inc., Wilmington, DE, USA). Extracted RNA was reverse transcribed to equal amounts of cDNA using the PrimeScript^TM^ reagent kit (Perfect Real Time) (Takara Bio Inc., Saint-Germain-en-Laye, France) according to the manufacturer’s protocol. The resulting cDNA was diluted ten-fold in Milli-Q water (Millipore).

### 4.4. Quantitative Real-Time PCR

The tissue and temporal distributions of *SchgrDib*, *SchgrSad* and *SchgrShd* were analyzed using quantitative (real-time) reverse transcription PCR (qRT-PCR) and the QuantStudio^®^ 3 Real-Time PCR Instrument (Applied Biosystems); 4 µL of cDNA was mixed with 5 µL Fast SYBR^®^ Green Master Mix (Applied Biosystems), 0.5 µL forward and 0.5 µL reverse primer (10 µM; primer sequences are listed in Table A1) and genes of interest were amplified under the following temperature conditions: 20 s at 95.0 °C and 40 cycles of 95 °C for 1 s and 60 °C for 20 s. Primers were validated and amplification specificity was determined as described by Lenaerts et al. (2017) [81]. Raw data were normalized according to the ddCt-method using a calibrator sample and two most stably expressed housekeeping genes, *ribosomal protein 49* (*RP49*) and *elongation factor 1α* (*EF1α*) for the tissue distribution and *β-actin* and *EF1α* for the temporal distribution. Each sample was measured in duplicate.

Knockdown efficiencies after RNAi-mediated knockdown of the ecdysteroid biosynthesis genes and transcript levels of genes of interest were analyzed in a similar way. In the experiment wherein *Halloween* genes were knocked down separately (See 2.6 RNA interference experiments), *glyceraldehyde 3-phosphate dehydrogenase* (*GAPDH*) and *CG13220* were selected as stably expressed housekeeping genes for all tissue samples. In the experiment wherein a combined knockdown of *Halloween* genes was performed, *EF1α* and *RP49* were selected.

### 4.5. Ecdysteroid Measurements Using an Enzyme Immunoassay

During the first reproductive cycle, hemolymph samples were collected every other day until day 18 of the adult stage and pooled in five groups of three locusts each. Locusts were pierced at their cuticle behind the hind leg and 10 µL of hemolymph was collected using a capillary and transferred to 270 µL of cold methanol (100%). Samples were stored at −20 °C until further processing. Ecdysteroid titers were measured in hemolymph samples using the enzyme immunoassay (EIA) as previously described by Marchal et al. (2011) [39]. In this assay, a rabbit L2 polyclonal antiserum with high affinity for ecdysone, 3-deoxyecdysone, 2-deoxyecdysone and a lower affinity (6- to 8-fold) for 20-hydroxyecdysone was used.

### 4.6. RNA Interference Experiments

**Production of dsRNA**. dsRNA constructs targeting *SchgrSpo* (*dsSpo*), *SchgrPhm* (*dsPhm*), *SchgrDib* (*dsDib*), *SchgrSad* (*dsSad*), *SchgrShd* (*dsShd*) and *GFP* (*dsGFP*) were synthesized using the MEGAscript^®^ RNAi Kit (Ambion) according to the manufacturer’s protocol. Since this kit makes use of a T7 RNA polymerase, genes of interest were amplified using T7 promoter flanked primers (see Table A2). Results were analyzed by horizontal 1% agarose gel electrophoresis and visualized with GelRedTM (Biotum) under UV light. The band of the expected size was excised and further purified using the GenElute™ Gel extraction Kit (Sigma-Aldrich Co., St. Louis, MO, USA). The resulting DNA fragment was cloned into a pCR4-TOPO vector using the TOPO^®^ TA Cloning Kit (Invitrogen), and after transformation of DH5α competent bacteria (*Escherichia coli*), and subsequent cloning, the plasmid DNA was purified using the GenElute^TM^ HP plasmid miniprep kit (Sigma Aldrich). The sequence of the insert was determined by Sanger sequencing to confirm its target specificity. When sequences were confirmed, dsRNA was produced using the T7 RNA polymerase and purified using a two-step purification protocol including a nuclease digestion and a column-based purification step. The purified dsRNA constructs were verified by horizontal 1% agarose gel electrophoresis and their concentrations were measured with a NanoDropTM 1000 Spectrophotometer.

**Knockdown of different *Halloween* genes separately.** Female locusts were collected on the day of molting to the fifth nymphal stage (N5D0) and injected with 5 µL of 80 ng/µL dsRNA, diluted in *S. gregaria* Ringer solution, targeting *SchgrSpo*, *SchgrPhm*, *SchgrDib*, *SchgrSad*, *SchgrShd* or *GFP* at day 6 (N5D6) of this last nymphal stage using a Hamilton syringe. Boost injections were given in the adult stage on day 1 (AdD1), day 5 (AdD5), day 9 (AdD9) and, in case of locusts destined for copulation behavior analysis, on day 13 (AdD13). On day 8 of the adult stage, female locusts were separated from male locusts and each knockdown condition was further divided into two different groups. One group of female locusts was dissected on day 12 of the adult stage (AdD12) to collect the ovaries and fat body for qRT-PCR analysis (*n* = 15 for all conditions) and to collect single ovarioles for staining the nuclei of the follicle epithelium cells (*n* = 4 for all conditions), while another group (*n* = 17, 20, 9, 13, 16, 19 for *dsGFP*-, *dsSpo*-, *dsPhm*-, *dsDib*-, *dsSad*- and *dsShd*-injected females, respectively) was kept alive for an observational analysis of their copulation behavior, as well as oviposition and hatching.

**Combined *Halloween* gene knockdown experiment**. In accordance with the previous knockdown experiment, female locusts were again collected on N5D0 and injected with dsRNA on N5D6. Boost injections were given on AdD1, AdD5, AdD9 and, in case of locusts destined for copulation behavior analysis, AdD13. In this experiment, *Halloween* genes *SchgrSpo*, *SchgrSad* and *SchgrShd* were targeted simultaneously by injecting 400 ng of each dsRNA construct (*dsSpo/Sad/Shd*) in each animal. Besides the control condition (*dsGFP*), a third condition targeting both *SchgrPhm* and *SchgrDib*, by injecting 400 ng of each dsRNA construct (*dsPhm/Dib*), was included. Female locusts were separated from male locusts on day 8 of the adult stage and divided into two different groups, one for tissue collection on AdD12, transcript analysis by qRT-PCR and visualization of the follicle epithelium cell nuclei (*n* = 15 and *n* = 4 for all three conditions, respectively) and one for observational analysis of copulation behavior, oviposition and hatching (*n* = 23, 21, 24 for *dsGFP*-, *dsSpo*/*Sad*/*Shd*- and *dsPhm*/*Dib*-injected females, respectively).

### 4.7. Oocyte Size, Copulation Behavior, Oviposition and Hatching

To determine the impact of RNAi-mediated knockdown of ecdysteroid biosynthesis enzymes on the female reproductive physiology, five individual oocytes were dissected from each female (fifteen females per condition) at an age of 12 days after the finale molt, of which the length and width were measured under a binocular microscope using a micrometer scale. Images of the ovaries and ovarioles were obtained using a light microscope (Zeiss SteREO Discovery. V8; Imaging sofware program Zen 2012, accessed from Leuven, Belgium) equipped with an AxioCam ICc3 camera using the AxioVision 4.7 (Carl Zeiss-Benelux, Oberkochen, Germany). Furthermore, using the calculated average length and width of the oocyte size measurements, an estimation of the oocyte’s volume was made by using the formula for calculating the volume of an ellipsoid body:(1)V=43πab2, with a=the average oocyte length andb=the average oocyte width

The processes of mating, oviposition and hatching were monitored and quantitatively analyzed. Locusts, which had molted to the adult stage on the same day, were kept in a gender-mixed group until day 8 of the adult stage. Males were then removed from the cages and females were transferred to individual containers. From day 9 of the adult stage on, each female locust was combined with a mature virgin male, which was at least one week older, for two hours. Mating was considered successful when there was a direct connection between the female and the male genitals. In this case, the male locust was allowed to stay with the female for another 22 h. In case no mating was observed, the male locust was removed from the container after two hours and the experiment was repeated the following day. After 24 h of mating, the male locusts were removed, and the female locusts received a pot filled with a moistened turf/sand (2:1) mixture, in which they could lay their eggs. These pots were controlled daily for presence of deposited egg batches and were replaced every four days. In case eggs were observed, the total number of eggs was counted for each individual female and the average egg length/width was measured as described above. In addition, an estimated volume of the eggs was calculated using Equation (1). All egg-containing pots were controlled daily for appearance of hatchlings. Finally, the number of hatchlings was counted and the percentage of hatching success (=100 × the number of hatchlings/number of eggs laid) was determined per condition. Appendix A shows a schematic overview of the experimental strategy that was followed in this study.

### 4.8. Visualisation of the Follicle Epithelium Cell Nuclei

On day 12 of the adult stage, single ovarioles were collected from the ovaries of *dsSpo*-, *dsPhm*-, *dsDib*-, *dsSad*-, *dsShd*-, *dsSpo*/*Sad*/*Shd*-, *dsPhm*/*Dib*- or *dsGFP*-treated females and freed from their surrounding membrane. The obtained ovarioles were fixated in 4% paraformaldehyde dissolved in PBS for 30 min at room temperature. After the fixation, ovarioles were washed for 2 times 5 min in PBS and incubated with 15 µg/mL 4′,6-diamidino-2-phenylindole (DAPI), which is known to bind to A-T rich regions of the DNA, for 10 min. Subsequently, ovarioles were washed in 0.2% Tween PBT and mounted on a glass slide using Mowiol solution (4 mM Mowiol^®^ 4-88, 65 M glycerol, finally diluted 1:3 in 0.2 M Tris pH = 8.5). Samples were stored in the dark at 4 °C and imaged using a confocal scanning microscope (FV1000-IX81, Olympus, Hamburg, Germany) and FluoViewer 4.2 software (Olympus accessed from Leuven, Belgium).

### 4.9. Statistical Analysis

Depending on the performed experiment, different statistical approaches were used. In the case of qRT-PCR analysis, transcript expression levels were normalized in Excel according to the ddCT-method using a calibrator sample and two selected most stably expressed housekeeping genes (see Section 2.4), which were selected using the geNorm software and the NormqPCR package in RStudio [82,83,84]. Afterwards, relative expression levels were converted to Graphpad Prism 6 (Graphpad Software Inc., San Diego, CA, USA) and log-transformed in order to create normalized data. Log-transformed data of the tissue and temporal distributions of *SchgrDib*, *SchgrSad* and *SchgrShd* were further analyzed using one-way ANOVA including the Tukey multiple comparisons test. Log-transformed, normalized expression levels obtained during the described knockdown experiments were always compared against the *dsGFP* conditions using a two-sided unpaired *t*-test, including the Welch’s correction when variances were significantly different from each other.

In the case of oocyte/egg length/width measurements and oocyte/egg volume estimations, the calculations were made in Excel and results were plotted using GraphPad Prism 6 (GraphPad Software Inc., San Diego, CA, USA) software. Normally distribution of the data was tested using the Shapiro–Wilk test. If the results were normally distributed, significant differences were determined using one-way ANOVA analysis with Dunnett’s multiple comparisons correction, if the results did not constitute a normal distribution, significant differences were determined using Kruskal–Wallis test with Dunn’s multiple comparisons correction. Furthermore, using the same software, the average length was plotted in function of the average width and simple linear regression was performed. Significant differences between the slopes and intercepts of the linear regression curves were determined using ANCOVA analysis.

Lastly, in the case of analyzing the effect of RNAi mediated knockdown on copulation behavior, oviposition and hatching, two different strategies were used. First, the cumulative percentages of females copulating and females laying eggs were represented by a survival curve. Differences between the different curves of the different knockdown conditions were determined using the Mantel–Cox test. Second, the number of days between egg copulation and egg laying, the number of laid eggs, the number of days between egg laying and hatching, the number of hatchlings and the hatchling success were represented as the mean ± S.E.M. The data were analyzed for a normal distribution using the Shapiro–Wilk test and in case of normally distributed data, significant differences were determined using one-way ANOVA analysis with Dunnett’s multiple comparisons correction. In case the results did not constitute a normal distribution, significant differences were determined using Kruskal–Wallis analysis with Dunn’s multiple comparisons correction.

## Figures and Tables

**Figure 1 ijms-23-09232-f001:**
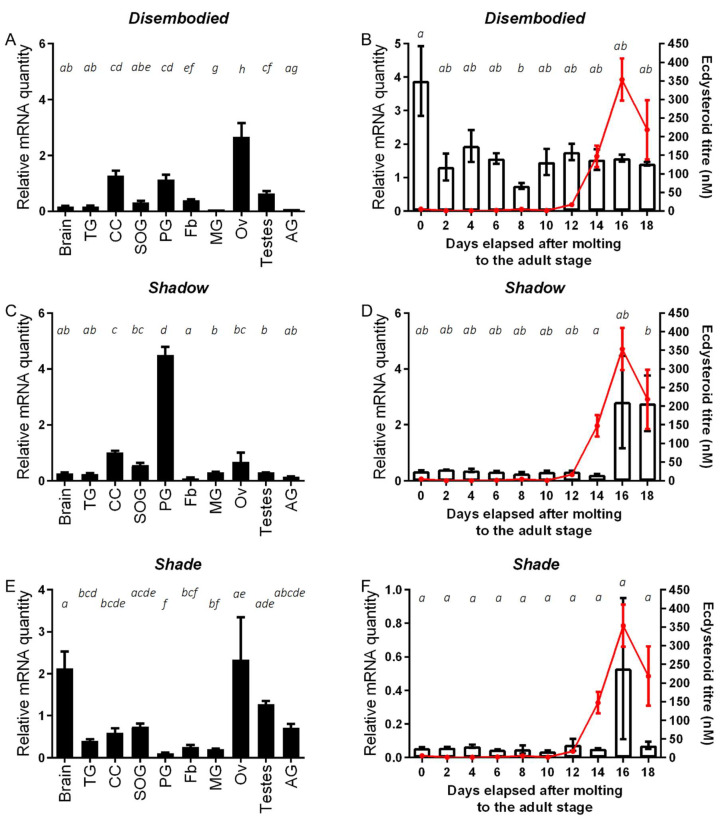
**Tissue and temporal distribution of *SchgrDib, SchgrSad* and *SchgrShd* transcripts**. Relative transcript levels of (**A**) *SchgrDib*, (**C**) *SchgrSad* and (**E**) *SchgrShd* measured in different tissues of adult locusts, using qRT-PCR. All tissues were dissected from adult female locusts 10 days after the final molt, except for the male accessory glands (AG) and testes. The data represent mean ± S.E.M. of three independent pools (10 animals/pool), run in duplicate and normalized to *ribosomal protein 49 (RP49)* and *elongation factor 1α* (*EF1α*) transcript levels. Other abbreviations on the X-axis: TG: thoracic ganglia; CC: *corpora cardiaca*; SOG: suboesophageal ganglion; PG: prothoracic glands; Fb: fat body; MG: midgut; Ov: ovaries. Temporal distribution profile of (**B**) *SchgrDib*, (**D**) *SchgrSad* and (**F**) *SchgrShd* in the ovaries during the first reproductive cycle. Using qRT-PCR, relative transcript levels of *SchgrDib*, *SchgrSad* and *SchgrShd* were measured every other day in the ovaries, starting on the day of molting to the adult stage (AdD0). Data represent mean ± S.E.M. (bars) of three independent pools of ten animals each, run in duplicate and normalized to *β-actin* and *EF1α* transcript levels. The ecdysteroid titer (red line), expressed in nM, throughout the first reproductive cycle was measured with an enzyme immunoassay (EIA). The data represent mean ± S.E.M. of 6–18 hemolymph samples taken from different animals per time point. Significant differences are indicated using letters as described by Piepho (2018), meaning that conditions having a letter in common do not significantly differ from each other (One-way ANOVA with Tukey’s multiple comparison test on log-transformed data) [46].

**Figure 2 ijms-23-09232-f002:**
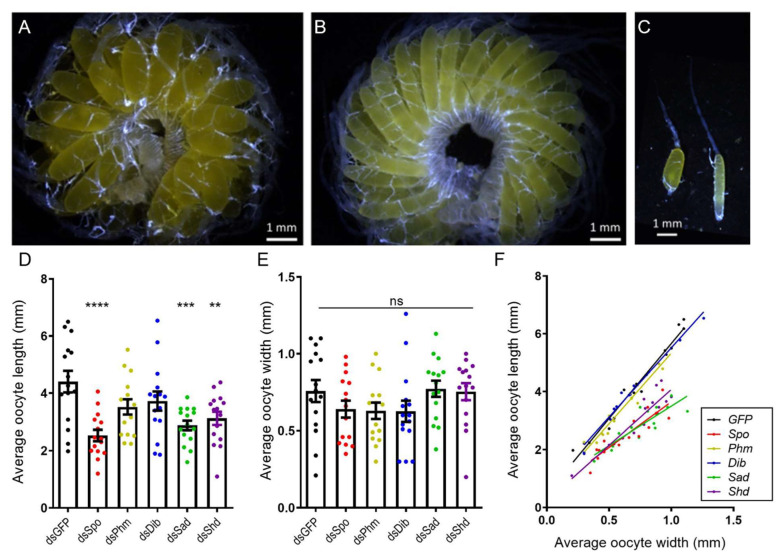
**Oocyte development in dsRNA-injected adult females twelve days after the final molt.** For each female locust, the width and length of five individual basal oocytes were measured twelve days after the final molt (AdD12). (**A**) Representative ovary for the observed phenotype upon RNAi-mediated knockdown of *SchgrSpo*, *SchgrSad* or *SchgrShd* (average size of basal oocytes: length = 2.84 ± 0.12 mm, width = 0.72 ± 0.03 mm). (**B**) Ovary representative for locusts in which *GFP*, *SchgrPhm* and *SchgrDib* were targeted (average size of basal oocytes: length = 3.88 ± 0.19 mm, width = 0.67 ± 0.04 mm). (**C**) Representative ovariole for the observed phenotype upon RNAi-mediated knockdown of *SchgrSpo*, *SchgrSad* or *SchgrShd* (left, basal oocyte: length = 2.4 mm, width = 0.9 mm) and the control (right, basal oocyte: length = 3.5 mm, width = 0.6 mm). Scale bars A-C = 1 mm. (**D**,**E**) The average length (**D**) or width (**E**) of five basal oocytes per locust and per condition, represented as mean ± S.E.M.; each dot represents one data point (*n* = 15). Significant differences are indicated by (an) asterisk(s) (**** *p* < 0.0001; *** *p* < 0.001 and ** *p* < 0.01; ns = ‘not significant’; One-way ANOVA with Dunnett’s Multiple Comparisons Test). (**F**) The data represent the average length (Y-axis) and width (X-axis) of five basal oocytes collected from each individual female treated with *dsSpo*, *dsPhm*, *dsDib*, *dsSad*, *dsShd* or *dsGFP* (coloured dots), as well as the linear regression (coloured lines) (*n* = 15 per condition). The Pearson‘s correlation coefficients for the *dsSpo*, *dsPhm*, *dsDib*, *dsSad*, *dsShd* or *dsGFP* knockdown conditions are 0.8954, 0.9771, 0.9888, 0.8587, 0.9205 and 0.9840, respectively (*p* < 0.0001 in all conditions). The linear regression curves of *dsSpo*-, *dsSad*- and *dsShd*- treated females are significantly different from the control condition (*p* = 0.00029 for *dsSpo*-treated females and *p* < 0.0001 for *dsSad*- and *dsShd*-treated females; ANCOVA).

**Figure 3 ijms-23-09232-f003:**
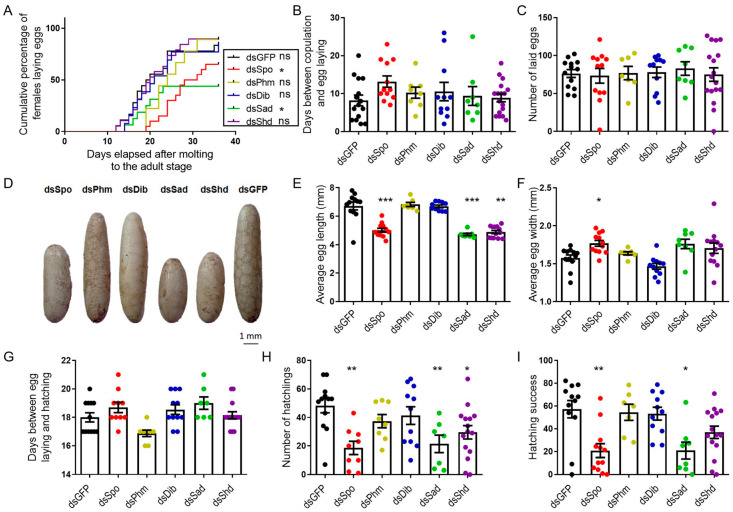
Effects of RNAi-mediated knockdown of the individual *Halloween* genes (*SchgrSpo*, *SchgrPhm*, *SchgrDib*, *SchgrSad* and *SchgrShd*) on egg size, egg laying and hatching of adult female *S. gregaria*. (**A**) Cumulative percentage of egg laying by dsRNA-injected female locusts: 65.00% (*n* = 12) for *dsSpo*-, 88.89% (*n* = 7) for *dsPhm*-, 84.62% (*n* = 11) for *dsDib*-, 50.00% (*n* = 8) for *dsSad*-, 89.47% (*n* = 13) for *dsShd*- and 83.33% (*n* = 12) for *dsGFP*-treated females. The curves of *dsSpo*- and *dsSad*-treated females significantly differ from the curve of *dsGFP*-treated females (*p* = 0.0335 and *p* = 0.0448, respectively; Mantel–Cox test) (* *p* < 0.05, ns = ‘not significant’). (**B**) The number of days between copulation and egg laying is presented as the mean ± S.E.M.; each dot represents one data point. (**C**) The number of eggs laid per condition is presented as the mean ± S.E.M.; each dot represents one data point. (**D**) Representative examples of eggs for each dsRNA treatment (*dsSpo*, *dsPhm*, *dsDib*, *dsSad*, *dsShd* or *dsGFP*). Scale bar = 1 mm. (**E**,**F**) The average length (**E**) or width (**F**) of 20 eggs per locust and per condition is presented as the mean ± S.E.M.; each dot represents one data point. Significant differences are indicated by (an) asterisk(s) (*** *p* < 0.001, ** *p* < 0.01, * *p* < 0.05; Kruskal–Wallis test with Dunn’s Multiple Comparison Test). (**G**) The number of days between egg laying and hatching is presented as the mean ± S.E.M.; each dot represents one data point. (**H**) The number of hatchlings per condition is presented as the mean ± S.E.M.; each dot represents one data point. Significant differences are indicated by (an) asterisk(s) (** *p* < 0.01, * *p* < 0.05; One-way ANOVA with Dunnett’s Multiple Comparison Test). (**I**) The hatching success (percentage of hatching eggs) is presented as the mean ± S.E.M.; each dot represents one data point. Significant differences are indicated by (an) asterisk(s) (** *p* < 0.01, * *p* < 0.05; Kruskal–Wallis test with Dunn’s Multiple Comparison Test).

**Figure 4 ijms-23-09232-f004:**
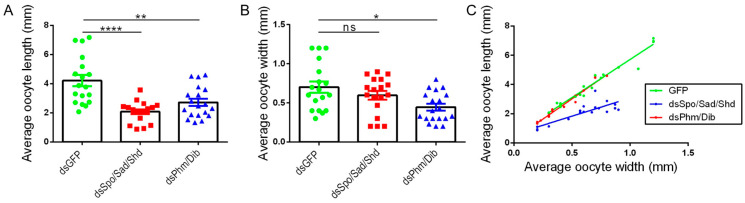
Oocyte development in adult females that were injected with a combination of *dsSpo*, *dsSad* and *dsShd* or with a combination of *dsPhm* and *dsDib*. (**A**,**B**) The average length (**A**) or the average width (**B**) of five basal oocytes per female locust twelve days after the final molt and per condition are represented as mean ± S.E.M.; each dot represents one data point (*n* = 15). Significant differences are indicated by (an) asterisk(s) (**** *p* < 0.0001; ** *p* < 0.01, * *p* < 0.05 and ns = ‘not significant’; Kruskal–Wallis test with Dunn’s Multiple Comparisons Test). (**C**) The data represent the average length (Y-axis) and width (X-axis) of five basal oocytes collected from *dsSpo*/*Sad*/*Shd*-, *dsPhm*/*Dib*-, or *dsGFP*-treated females twelve days after the final molt (coloured dots), as well as the linear regression (coloured lines) (*n* = 15 per condition). The Pearson‘s correlation coefficients for the *dsSpo*/*Sad*/*Shd*, *dsPhm*/*Dib* and *dsGFP* knockdown conditions are 0.8276, 0.9824 and 0.9778, respectively (*p* < 0.0001 in all conditions). The linear regression curve of *dsSpo*/*Sad*/*Shd* is significantly different from the control condition (*p* < 0.0001; ANCOVA).

**Figure 5 ijms-23-09232-f005:**
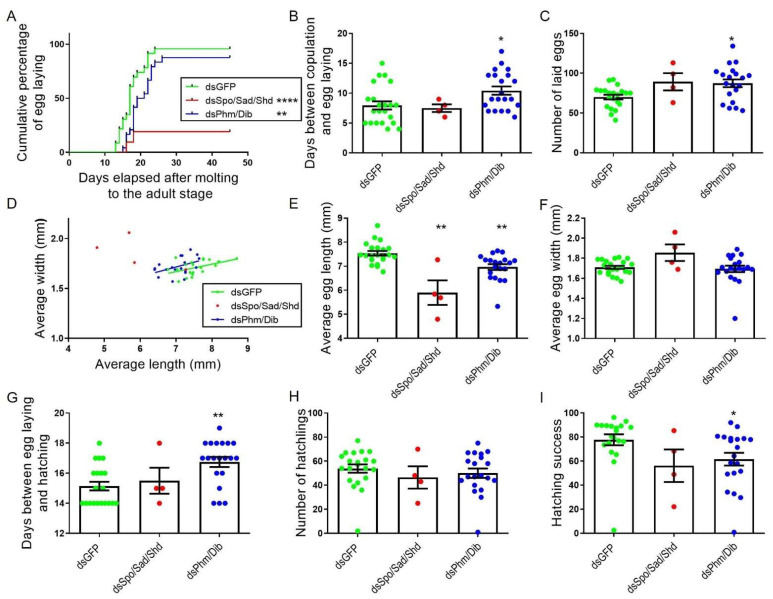
Effects of RNAi-mediated knockdown of *SchgrSpo*/*SchgrSad*/*SchgrShd* and *SchgrPhm*/*SchgrDib* on egg size, egg laying and hatching of female *S. gregaria*. (**A**) Cumulative percentage of egg laying by dsRNA-injected female locusts: 19.05% for *dsSpo*/*Sad*/*Shd*- (*n* = 4), 87.5% for *dsPhm*/*Dib*- (*n* = 24) and 95.65% for *dsGFP*-treated (*n* = 23) females. The curves of *dsSpo*/*Sad*/*Shd* and *dsPhm*/*Dib*-treated females significantly differ from the curve of *dsGFP*-treated females (**** *p* < 0.0001 for *dsSpo*/*Sad*/*Shd* and ** *p* < 0.01 for *dsPhm*/*Dib*; Mantel–Cox test). (**B**) The number of days between copulation and egg laying is presented as the mean ± S.E.M.; each dot represents one data point. Significant differences are indicated by (an) asterisk(s) (* *p* < 0.05; Kruskal–Wallis test with Dunn’s Multiple Comparisons test). (**C**) The number of eggs laid per condition is presented as the mean ± S.E.M.; each dot represents one data point. Significant differences are indicated by (an) asterisk(s) (* *p* < 0.05; Kruskal–Wallis test with Dunn’s Multiple Comparisons test). (**D**) The data represent the average length (Y-axis) and width (X-axis) of eggs laid by *dsSpo*/*Sad*/*Shd*-, *dsPhm*/*Dib*- or *dsGFP*-treated females (coloured dots), as well as the linear regression (coloured lines) (*dsSpo*/*Sad*/*Shd n* = 4, *dsPhm*/*Dib n* = 20 and *dsGFP n* = 21). The Pearson‘s correlation coefficients for the *dsSpo*/*Sad*/*Shd*, *dsPhm*/*Dib* and *dsGFP* knockdown conditions are −0.6416 (*p* = 0.3584), 0.4309 (*p* = 0.0655) and 0.4624 (*p* = 0.0348), respectively. (**E**) The average length of seven eggs per locust is presented as the mean ± S.E.M.; each dot represents one data point. Significant differences are indicated by (an) asterisk(s) (** *p* < 0.01; Kruskal–Wallis test with Dunn’s Multiple Comparisons test). (**F**) The average width of seven eggs per locust is presented as the mean ± S.E.M.; each dot represents one data point. (**G**) The number of days between egg laying and hatching is presented as the mean ± S.E.M.; each dot represents one data point. Significant differences are indicated by (an) asterisk(s) (** *p* < 0.01; Kruskal–Wallis test with Dunn’s Multiple Comparisons test). (**H**) The number of hatchlings per condition is presented as the mean ± S.E.M.; each dot represents one data point. (**I**) The hatching success percentage is presented as the mean ± S.E.M.; each dot represents one data point. Significant differences are indicated by (an) asterisk(s) (* *p* < 0.05; Kruskal–Wallis test with Dunn’s Multiple Comparisons test).

## Data Availability

The raw data incorporated in the figures can be made available upon request to the corresponding author.

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
