# Peer review of "Knockdown of the Halloween Genes spook, shadow and shade Influences Oocyte Development, Egg Shape, Oviposition and Hatching in the Desert Locust"

_ijms, 2022, doi:10.3390/ijms23169232_

Round 1

Reviewer 1 Report

The manuscript entitled “Knockdown of the Halloween genes spook, shadow and shade influences oocyte development, egg shape, oviposition and hatching in the desert locust” deals with the separate or combinatorial silencing of functional genes of Schistocerca gregaria demonstrating their role in oocyte development, oviposition and hatching of the eggs. The manuscript is very well written, clearly organized and technically sound. Huge amount of work has been carried out to describe a picture of cross-talk among these genes. Figures and conclusions are well described and supported by statistical analyses. Few minor comments are listed below.

- Symbols of statistical significance are missing in Figure 1 despite they are cited in Lines 149-151.

- Letters of panels (A to F) are also missing in Figure 1.

- Explicit abbreviation EIA at line 148.

- Explicit abbreviation at lines 156-157 anticipating description presented at lines 623 in Mat and Met.

- In fig. S1 only one primer is visible in blue, but not the other of the pair for dsRNA construct.

- Caption to fig. S5 is entitled “Supplementary Figure S1”. Please amend it.    

- In combined silencing experiment (dsSpo/Sad/Shd) a significant reduction of target transcript was measured for SchgrShd and not for the other two genes (fig. S12). Please discuss it.

Author Response

We are very grateful to the editor(s) and reviewers for their constructive comments, which we have now taken into account when revising our manuscript, as outlined below:

The manuscript entitled “Knockdown of the Halloween genes spook, shadow and shade influences oocyte development, egg shape, oviposition and hatching in the desert locust” deals with the separate or combinatorial silencing of functional genes of Schistocerca gregaria demonstrating their role in oocyte development, oviposition and hatching of the eggs. The manuscript is very well written, clearly organized and technically sound. Huge amount of work has been carried out to describe a picture of cross-talk among these genes. Figures and conclusions are well described and supported by statistical analyses. Few minor comments are listed below.

We thank the reviewer for these positive comments and for appreciating our manuscript, which indeed represents a huge amount of work. We greatly appreciate that the reviewer is recognizing this. We are also very grateful to the reviewer for his/her constructive remarks that helped us a lot in improving our manuscript.

- Symbols of statistical significance are missing in Figure 1 despite they are cited in Lines 149-151.

This is adjusted in the revised manuscript.

- Letters of panels (A to F) are also missing in Figure 1.

This is adjusted in the revised manuscript.

- Explicit abbreviation EIA at line 148.

This is adjusted in the revised manuscript.

- Explicit abbreviation at lines 156-157 anticipating description presented at lines 623 in Mat and Met.

This is adjusted in the revised manuscript.

- In fig. S1 only one primer is visible in blue, but not the other of the pair for dsRNA construct.

This is adjusted in the revised manuscript.

- Caption to fig. S5 is entitled “Supplementary Figure S1”. Please amend it.   

This is adjusted in the revised manuscript.

- In combined silencing experiment (dsSpo/Sad/Shd) a significant reduction of target transcript was measured for SchgrShd and not for the other two genes (fig. S12). Please discuss it.

Indeed, although more severe phenotypic effects were observed in the combined silencing experiment when compared to the individual gene knockdown analyses, no significant effects on the transcript levels of SchgrSpo and SchgrSad were observed (in contrast to the separate gene knockdowns). However, sampling of tissues for qPCR implies that animals have to be sacrificed at a certain timepoint and it is impossible to test all possible timepoints. Since there are at least two weeks between the first dsRNA injection and dissection of the animals, it could be suggested that compensatory effects, which are not unusual for Halloween gene knockdowns (Kumar et al., 2017; Marchal et al., 2011; Rossi et al., 2015; Zhang et al., 2018), may have taken place, but insufficient to revert the phenotype.

Bibliography:

Kumar, R., Mota, L. C., Litoff, E. J., Rooney, J. P., Boswell, W. T., Courter, E., Henderson, C. M., Hernandez, J. P., Corton, J. C., Moore, D. D., & Baldwin, W. S. (2017). Compensatory changes in CYP expression in three different toxicology mouse models: CAR-null, Cyp3a-null, and Cyp2b9/10/13-null mice. PLOS ONE, 12(3), e0174355. https://doi.org/10.1371/journal.pone.0174355

Marchal, E., Badisco, L., Verlinden, H., Vandersmissen, T., van Soest, S., van Wielendaele, P., & vanden Broeck, J. (2011). Role of the Halloween genes, Spook and Phantom in ecdysteroidogenesis in the desert locust, Schistocerca gregaria. Journal of Insect Physiology, 57(9), 1240–1248. https://doi.org/10.1016/j.jinsphys.2011.05.009

Rossi, A., Kontarakis, Z., Gerri, C., Nolte, H., Hölper, S., Krüger, M., & Stainier, D. Y. R. (2015). Genetic compensation induced by deleterious mutations but not gene knockdowns. Nature, 524(7564), 230–233. https://doi.org/10.1038/nature14580

Zhang, W., Chen, W., Li, Z., Ma, L., Yu, J., Wang, H., Liu, Z., & Xu, B. (2018). Identification and Characterization of Three New Cytochrome P450 Genes and the Use of RNA Interference to Evaluate Their Roles in Antioxidant Defense in Apis cerana cerana Fabricius. Frontiers in Physiology, 9(NOV). https://doi.org/10.3389/fphys.2018.01608

Reviewer 2 Report

Schellens et al. reported the functional characterization of the three Halloween genes, spook, shadow, and shade in the desert locust, Schistocerca gregaria. They first cloned the responsible genes in this insect species, analyzed the gene expression pattern, and then conducted RNAi-mediated knockdown experiments to analyze the effects on oocyte development and reproduction. They found that knockdown of Spo, Sad, and Shd reduced oocyte/egg length and hatching rate. While JH is known to be a crucial factor for oocyte development, they showed that the knockdown of the Halloween genes did not affect JH signaling as revealed by the expression levels of JH-target genes in the fat body and ovaries. In conclusion, this manuscript will advance our understanding of the role of Halloween genes in oocyte development in the desert locust.

Major comments.

I would really appreciate it if the authors experimentally show the ecdysteroid levels in the knockdown ovary and laid eggs. This information will help to understand the oogenesis defects and embryonic lethal phenotype and further strengthen the manuscript.

It would be great if the authors show the oocyte/egg volume in addition to the length/width data. These are simply specified as an ellipsoid body.

Related to Figures 3H/I, do Spo, Sad, Shd knockdown eggs show any abnormalities, like the Halloween phenotype observed in the fruit fly?

Minor comments.

In the introduction (lines 46-), they mentioned how 20E functions as a transcriptional regulator, but these are too simplified. 20E target genes significantly vary depending on tissues and stages. Moreover, EcR acts as a transcriptional activator and suppressor in a ligand-dependent/independent manner. To avoid confusion, the authors should consider these facts.

It is unclear why the authors conducted triple/double knockdown experiments of the Halloween genes. Please clarify the rational reason. A related issue is why the triple knockdown did not show lethality despite every single knockdown show lethality (Fig 5H versus Fig 3H).

Please specify which ecdysteroid molecules were actually measured in the EIA. This can be dependent on the specificity of the antibody used in the assay.

Supplementary Figure S1 should be S5 (line 71).

Author Response

We are very grateful to the editor(s) and reviewers for their constructive comments, which we have now taken into account when revising our manuscript, as outlined below:

Schellens et al. reported the functional characterization of the three Halloween genes, spook, shadow, and shade in the desert locust, Schistocerca gregaria. They first cloned the responsible genes in this insect species, analyzed the gene expression pattern, and then conducted RNAi-mediated knockdown experiments to analyze the effects on oocyte development and reproduction. They found that knockdown of Spo, Sad, and Shd reduced oocyte/egg length and hatching rate. While JH is known to be a crucial factor for oocyte development, they showed that the knockdown of the Halloween genes did not affect JH signaling as revealed by the expression levels of JH-target genes in the fat body and ovaries. In conclusion, this manuscript will advance our understanding of the role of Halloween genes in oocyte development in the desert locust.

We thank the reviewer for these constructive comments and for recognizing the impact of our study on understanding the role of the Halloween genes in oocyte development in the desert locust. We are very grateful for the different suggestions that helped us in adapting and improving our manuscript.

Major comments.

I would really appreciate it if the authors experimentally show the ecdysteroid levels in the knockdown ovary and laid eggs. This information will help to understand the oogenesis defects and embryonic lethal phenotype and further strengthen the manuscript.

Previously published results have shown that knocking down Halloween genes such as spook, phantom, shadow and shade indeed resulted in significantly reduced ecdysteroid levels, fully in line with their role in ecdysteroid synthesis (Marchal et al., 2011, 2012; Peng et al., 2019; Sugahara et al., 2017; Zhou et al., 2020). Furthermore, during a first preliminary experiment in which the different Halloween genes were silenced at once, the ecdysteroid content was measured after the dsRNA treatments, showing that knockdown significantly lowers the ecdysteroid levels in the ovaries of knockdown locusts, when compared to control locusts. These experiments were not repeated for the separate knockdowns considering the high cost of the commercial kits and the number of conditions and replicates. Moreover, for analyzing ecdysteroid levels in ovaries, animals needed to be sacrificed and would then not have been utilized anymore for analyzing mating, oviposition, egg size/shape measurements, hatching, etc.

It would be great if the authors show the oocyte/egg volume in addition to the length/width data. These are simply specified as an ellipsoid body.

In our revised manuscript, we have added an extra supplementary figure (S9) which shows the oocyte/egg volume in the different conditions. Also the M&M, results and discussion sections are adapted accordingly.

Related to Figures 3H/I, do Spo, Sad, Shd knockdown eggs show any abnormalities, like the Halloween phenotype observed in the fruit fly?

When counting and measuring the eggs in the different conditions, no obvious abnormalities, apart from the spherical shape, were observed. Hatchlings obtained from the different treatment conditions were further observed for at least two weeks, but no abnormal phenotypes or moulting defects were observed. However, in the dsSpo, dsSad and dsShd knockdown conditions, respectively, only ca. 20%, 20% and 40% of the eggs hatched successfully, which may point at in ovo developmental defects.

Minor comments.

In the introduction (lines 46-), they mentioned how 20E functions as a transcriptional regulator, but these are too simplified. 20E target genes significantly vary depending on tissues and stages. Moreover, EcR acts as a transcriptional activator and suppressor in a ligand-dependent/independent manner. To avoid confusion, the authors should consider these facts.

We thank the reviewer for this suggestion and updated the paragraph accordingly.

It is unclear why the authors conducted triple/double knockdown experiments of the Halloween genes. Please clarify the rational reason. A related issue is why the triple knockdown did not show lethality despite every single knockdown show lethality (Fig 5H versus Fig 3H).

These two combinations were selected based on the results we obtained in the separate Halloween gene knockdown experiments. As similar phenotypes were observed after downregulating spook, shadow or shade, we decided to combine these three genes in one single knockdown experiment to determine if an even more severe effect would appear. Since downregulation of phantom or disembodied did not reveal the same phenotype as observed after silencing of spook, shadow or shade, we were wondering if this phenotype would perhaps appear in a stronger (combined) knockdown condition, and this was not the case. We apologize if this rationale was not clear enough and have now explained this more clearly in the revised manuscript.

In the case of the triple knockdown (Fig 5H), only 4 females were able to lay eggs, whereas all other females never laid eggs and died during the period of egg laying of the control females. Therefore, it could be reasoned that those 4 females only had a partial depletion of spook, shadow and shade as this is inherent to knockdown strategies. Furthermore, since we only have 4 measurements for this condition, it is difficult to have statistically significant results. Hence, as a consequence of this partial knockdown and the limited number of replicates, we did not observe a significant effect on the number of hatchlings.

Please specify which ecdysteroid molecules were actually measured in the EIA. This can be dependent on the specificity of the antibody used in the assay.

As described by Marchal et al. (2011), the performed EIA is based on rabbit L2 polyclonal antibodies that have a strong affinity for ecdysone, 3-deoxyecdysone, 2-deoxyecdysone and a 6- 8-fold lower affinity for 20E. We updated our M&M section to provide this information.

Supplementary Figure S1 should be S5 (line 71).

This was adapted in the revised manuscript.

References

Marchal, E., Badisco, L., Verlinden, H., Vandersmissen, T., van Soest, S., van Wielendaele, P., & vanden Broeck, J. (2011). Role of the Halloween genes, Spook and Phantom in ecdysteroidogenesis in the desert locust, Schistocerca gregaria. Journal of Insect Physiology, 57(9), 1240–1248. https://doi.org/10.1016/j.jinsphys.2011.05.009

Marchal, E., Verlinden, H., Badisco, L., Van Wielendaele, P., & Vanden Broeck, J. (2012). RNAi-mediated knockdown of Shade negatively affects ecdysone-20-hydroxylation in the desert locust, Schistocerca gregaria. Journal of Insect Physiology, 58(7), 890–896. https://doi.org/10.1016/j.jinsphys.2012.03.013

Peng, L., Wang, L., Zou, M.-M., Vasseur, L., Chu, L.-N., Qin, Y.-D., Zhai, Y.-L., & You, M.-S. (2019). Identification of Halloween Genes and RNA Interference-Mediated Functional Characterization of a Halloween Gene shadow in Plutella xylostella. Frontiers in Physiology, 10(AUG). https://doi.org/10.3389/fphys.2019.01120

Sugahara, R., Tanaka, S., & Shiotsuki, T. (2017). RNAi-mediated knockdown of SPOOK reduces ecdysteroid titers and causes precocious metamorphosis in the desert locust Schistocerca gregaria. Developmental Biology, 429(1), 71–80. https://doi.org/10.1016/j.ydbio.2017.07.007

Zhou, X., Ye, Y.-Z., Ogihara, M. H., Takeshima, M., Fujinaga, D., Liu, C.-W., Zhu, Z., Kataoka, H., & Bao, Y.-Y. (2020). Functional analysis of ecdysteroid biosynthetic enzymes of the rice planthopper, Nilaparvata lugens. Insect Biochemistry and Molecular Biology, 123(April). https://doi.org/10.1016/j.ibmb.2020.103428

This manuscript is a resubmission of an earlier submission. The following is a list of the peer review reports and author responses from that submission.

Round 1

Reviewer 1 Report

In this paper, Schellens et al. report that knockdown of Halloween genes Spo, Sad and Shd but not Phm and Dib resulted in blocked oocyte development as well as reduced oviposition and egg hatching rate in the desert locust, Schistocerca gregaria. The reduced size and altered shape of basal oocytes and deposited were also observed. The authors conclude that Halloween genes Spo, Sad and Shd determine oocyte length as stated in the title. Data obtained in this study provide evidence that Spo, Sad and Shd play an important role in desert locust oogenesis and fecundity. However, these data are insufficient to support the conclusion that Spo, Sad and Shd determine oocyte length. The shortened basal oocytes and eggs could be an indirect result from reduced Vg accumulation or inhibited development of follicular epithelium. These points should be clarified with additional experiments. Otherwise, the title and conclusion on oocyte/egg length need revising.

  1. Result 3.2 and Fig 1 show the spatiotemporal expression pattern of SchgrDib, SchgrSad and SchgrShd. For a better understanding of Halloween genes in regulating locust oogenesis, the spatiotemporal expression patterns of SchgrSpo and SchgrPhm in the first gonadotrophic cycle could be included.
  2. In Fig 2 and Materials and Methods, please provide the number of examined basal oocytes. Fig 2A-D show that the oocytes of Spo-, Sad- or Shd-depleted females had shorter basal oocytes and abnormal spherical shape compared to the dsGFP controls. It is highly recommended to examine the change of follicle epithelium and follicle cell number.
  3. In addition to the mRNA levels of SchgrVg1 and SchgrVg2, their protein levels in the fat body, hemolymph and ovary should be measured after Spo, Sad and Shd RNAi. It is necessary to help understand whether the altered shape and size of basal oocyte were due to less deposition of Vg.
  4. The representative defective phenotypes (images) of ovary, ovarioles, basal oocytes and eggs resulted from simultaneous knockdown of Spo, Sad and Shd should be included and compared in parallel with that caused by individual knockdown of these genes.

Reviewer 2 Report

The manuscript entitled “Halloween genes spook, shadow and shade determine oocyte length in the desert locust, Schistocerca gregaria” deals with the separate or combinatorial silencing of functional genes of S. gregaria demonstrating their role in oocyte development, oviposition and hatching of the eggs. The manuscript is very well written, clearly organized and technically sound. Huge amount of work has been carried out to describe a picture of cross-talk among these genes. Figures and conclusions are well described and supported by statistical analyses. Few major comments are listed below.

- The manuscript title could be more general to comprise all the results on oocyte development. Combinatorial silencing ended up in a huge effect on egg laying (Fig. 5A) that is a very interesting result not included in the present title.

- Statistical analyses are missing in Figure 1 and therefore sentences at lines 265-267, 274-279, 512-514, 525-528 are not very well supported. Statistical comparisons also for these data surely will improve the manuscript.

- Paragraph 3.4.1 is hard to read. It could be shorten up by stressing the point on significant differences and avoid providing not significant statistical values.

- In combined silencing experiment (dsSpo/Sad/Shd) a significant reduction of target transcript was measured for SchgrShd and not for the other two genes (fig. S9). Please discuss it.

Minor comments

L119: list here all the four species used for alignment and insert reference to pertinent Supplementary figures.

 Fig. S7 and S10 have low quality resolution.

Please add a paragraph in mat and meth section to explain statistical tests and software details for analyses and graphs. Move into this new paragraph the sentence now at lines 159-162. Statistic has been used not only for gene expression, but also for all other morphometric analyses, it would be better to describe all the tests used in a unique section.

 Reduce as much as possible abbreviations in captions in order they result self-explaining.

In the end a curiosity. Is it feasible to administer dsRNAs and measure ecdysteroid titer in silenced insects? Did you try something like that? Any clues?

Reviewer 3 Report

In this study, the authors reported the temporal-spatial expression profiles of 5 genes involved in 20E synthesis pathway and their function in reproduction in the desert locust, Schistocerca gregaria. The description of experimental design and methods were clear and in detail. However, some experiments are unreasonable. (1) Why dsRNA was injected at 5th instar locust, but not at 0 day of adult emergence? Whether cross-developmental stages injection affect the moulting process to adult? (2) Why only spo/sad/shd or phm/dib combination RNAi was used? But not other combination? (3) Why only detect the 20E titers in hemolymph? In adults, the reproductive organ may reproduce 20E and function locally. The 20E titer of ovary should be detected. (4) The observations of oocytes by confocal microscopy were necessary because the oocyte or egg phenotypes were probably the cause of follicle cells dysfunction by Halloween genes RNAi.